# Survey on Carbapenem-Resistant Bacteria in Pigs at Slaughter and Comparison with Human Clinical Isolates in Italy

**DOI:** 10.3390/antibiotics11060777

**Published:** 2022-06-07

**Authors:** Silvia Bonardi, Clotilde Silvia Cabassi, Gerardo Manfreda, Antonio Parisi, Enrico Fiaccadori, Alice Sabatino, Sandro Cavirani, Cristina Bacci, Martina Rega, Costanza Spadini, Mattia Iannarelli, Cecilia Crippa, Ferdinando Ruocco, Frédérique Pasquali

**Affiliations:** 1Department of Veterinary Science, University of Parma, 43126 Parma, Italy; clotildesilvia.cabassi@unipr.it (C.S.C.); sandro.cavirani@unipr.it (S.C.); cristina.bacci@unipr.it (C.B.); martina.rega@unipr.it (M.R.); costanza.spadini@unipr.it (C.S.); iannarelli.mattia@gmail.com (M.I.); 2Food Safety Unit, Department of Agricultural and Food Sciences Alma Mater Studiorum, University of Bologna, 40064 Ozzano dell’Emilia, Italy; gerardo.manfreda@unibo.it (G.M.); cecilia.crippa2@unibo.it (C.C.); frederique.pasquali@unibo.it (F.P.); 3Istituto Zooprofilattico Sperimentale della Puglia e della Basilicata, 70017 Putignano, Italy; antonio.parisi@izspb.it; 4Nephrology Unit, Parma University-Hospital, Department of Medicine and Surgery, Parma University, 43126 Parma, Italy; enrico.fiaccadori@unipr.it (E.F.); alice.sabatino86@gmail.com (A.S.); 5National Veterinary Service, 43126 Parma, Italy; fruocco@ausl.it

**Keywords:** carbapenem resistance, ESBLs, OXA genes, antimicrobial resistance, *Pseudomonas aeruginosa*, pigs

## Abstract

This study is focused on resistance to carbapenems and third-generation cephalosporins in Gram-negative microorganisms isolated from swine, whose transmission to humans via pork consumption cannot be excluded. In addition, the common carriage of carbapenem-resistant (CR) bacteria between humans and pigs was evaluated. Sampling involved 300 faecal samples collected from slaughtered pigs and 300 urine samples collected from 187 hospitalised patients in Parma Province (Italy). In swine, MIC testing confirmed resistance to meropenem for isolates of *Pseudomonas aeruginosa* and *Pseudomonas oryzihabitans* and resistance to cefotaxime and ceftazidime for *Escherichia coli*, *Ewingella americana*, *Enterobacter agglomerans*, and *Citrobacter freundii*. For *Acinetobacter lwoffii*, *Aeromonas hydrofila, Burkolderia cepacia*, *Corynebacterium indologenes*, *Flavobacterium odoratum*, and *Stenotrophomonas maltophilia*, no EUCAST MIC breakpoints were available. However, ESBL genes (*bla*_CTXM-1_, *bla*_CTX-M-2_, *bla*_TEM-1_, and *bla*_SHV_) and AmpC genes (*bla*_CIT_, *bla*_ACC_, and *bla*_EBC_) were found in 38 and 16 isolates, respectively. *P. aeruginosa* was the only CR species shared by pigs (4/300 pigs; 1.3%) and patients (2/187; 1.1%). *P. aeruginosa* ST938 carrying *bla*_PAO_ and *bla*_OXA396_ was detected in one pig as well as an 83-year-old patient. Although no direct epidemiological link was demonstrable, SNP calling and cgMLST showed a genetic relationship of the isolates (86 SNPs and 661 allele difference), thus suggesting possible circulation of CR bacteria between swine and humans.

## 1. Introduction

Antimicrobial resistance (AMR) represents an important field of One Health, a concept based on the definition of communication spaces among different environments and hosts, such as humans, animals, plants, water, and soil [1]. Among antimicrobials, resistance to carbapenems is of great concern since they are the “last-line defence” to treat human infections by multiresistant Gram-negative bacilli [2,3,4,5]. Carbapenems are not licenced for use in food-producing animals in the EU [6], but other β-lactams are commonly used, such as extended-spectrum cephalosporins, which could provide a selection pressure favouring the expression of carbapenem resistance (CR) [7]. In recent decades, intensive farming has been frequently associated with the use/abuse of antimicrobials, and AMR is common among microorganisms isolated from food-producing animals, with subsequent possible transmission to humans via direct contact with animals or ingestion of derived food products [8]. Furthermore, carbapenem-resistant (CR) and carbapenemase-producing (CP) microorganisms of human origin could be transmitted to livestock from environmental sources (e.g., hospital sewage and wastewater treatment plants contaminating water and soil) [9,10,11].

CR bacteria may deactivate the carbapenems through two main mechanisms: (i) acquisition of genes that encode for enzymes capable of hydrolysing the carbapenems, called carbapenemases; (ii) reduction in the accumulation of antibiotics due to porin deficiency combined with expression of β-lactamases with a poor affinity for carbapenems, as AmpC-β-lactamases or extended-spectrum β-lactamases (ESBLs) [4,12]. Different classes of carbapenemases are found in CP microorganisms. Class A KPC enzymes can hydrolyse all β-lactams in use but ceftaroline/avibactam [13]. Class B enzymes belonging to the IMP, VIM, and NDM families show the highest carbapenemase activity [2]. Class D carbapenem-hydrolysing oxacillinases (OXAs) have emerged in several bacterial species as responsible for carbapenem resistance [14].

In pigs, CR was observed in microorganisms belonging to different bacterial species, such as *Escherichia coli* harbouring *bla*_VIM-1_ [15], *bla*_OXA-181_ [16], or *bla*_IMP-27_ [7], and *Salmonella* Infantis harbouring *bla*_VIM-1_ [17]. The gene *bla*_IMP-27_ was identified in microorganisms isolated from swine faecal and environmental samples, such as *Klebsiella oxytoca, Proteus mirabilis, Proteus vulgaris*, *Proteus rettgeri*, *Enterobacter cloacae*, and *Morganella morganii* [7]. Despite these interesting findings assessing the dissemination of carbapenemases to microorganisms detected in swine, further studies are needed to understand the role of swine in the epidemiology of CR Gram-negative infections in humans.

CR may also result from the synthesis of ESBL and/or AmpC β-lactamase enzymes in association with quantitative and/or qualitative deficiency in the expression of outer membrane porins [18]. ESBLs are inhibitor-susceptible β-lactamases that hydrolyse penicillins, cephalosporins, and aztreonam but show a weak affinity for carbapenems. They are encoded by mobile genes, mostly belonging to SHV, TEM, and CTX m families [19]. Notably, the CTX m gene family is the most frequently detected in bacteria carried by food-producing animals and humans [20]. ESBLs, in addition to carbapenemases, have compromised the activity of almost all penicillins and cephalosporins, leading to the development of combination therapy with other β-lactams, β-lactamase inhibitors, or antibiotics from other classes [21].

Among cephalosporinases, transmissible plasmid-mediated AmpC β-lactamases commonly hydrolyse narrow-, broad-, and extended-spectrum cephalosporins and cephamycins and resist inhibition by clavulanate, sulbactam, and tazobactam. The most common plasmid-mediated AmpC β-lactamases belong to the CMY, FOX, and DHA families. These enzymes have been detected in different microorganisms, such as *Klebsiella* spp., *Salmonella* spp., *Citrobacter freundii*, *Enterobacter aerogenes*, *P. mirabilis*, and *E. coli* [22], which are often classified as multidrug-resistant [19].

Since a decade ago, resistance to third- and fourth-generation cephalosporins in ESBL/AmpC-producing *E. coli* of animal origin has been showing an increasing trend in the European countries, probably due to the antimicrobial treatments in livestock. Their transmission to humans poses a threat, and a worrying association of ESBL/AmpC-producing *E. coli* strains between fattening pigs and pig farmers has been assessed [23,24,25]. Given the importance of cephalosporins in human medicine and the emergence and distribution of resistant bacteria, monitoring of ESBL/AmpC-producing *E. coli* from food-producing animals and food thereof has been mandatory in the EU since 2014 [26]. Such monitoring is of crucial importance to human health because the expression of ESBL/AmpC genes in microorganisms showing porin deficiencies may be related to resistance to carbapenems [18].

The main objective of this study was focused on the One Health concept that CR bacteria could be shared between the animal and human compartments. In this perspective, meropenem was the carbapenem selected for the antimicrobial sensitivity testing because of its potency and wide antimicrobial spectrum, including Gram-negative and Gram-positive bacteria [27,28,29]. Globally, the aims of the study were (i) the detection of CR microorganisms in faecal samples collected from pigs at slaughter; (ii) the evaluation of their phenotypical resistance to carbapenems; (iii) the distribution of carbapenemase-producing genes in the porcine isolates; (iv) the genomic comparison of CR porcine microorganisms with CR bacteria isolated from hospitalised patients in the same geographical area of northern Italy. In addition, (v) resistance to third-generation cephalosporins (cefotaxime and ceftazidime) was tested, followed by detection of ESBL/AmpC genes which can contribute to CR in microbial isolates.

## 2. Results

### 2.1. Phenotypical Antimicrobial Resistance in Pig Isolates

A total of 38 Gram-negative isolates from pig faeces were selected based on the Kirby–Bauer test. They belonged to different species—namely, *Acinetobacter lwoffii* (*n* = 1), *Aeromonas hydrophila* (*n* = 2), *Burkolderia cepacia* (*n* = 3), *Citrobacter freundii* (*n* = 1), *Cryseobacterium indologenes* (*n* = 1), *Enterobacter agglomerans* (*n* = 1), *Escherichia coli* (*n* = 4), *Ewingella americana* (*n* = 1), *Flavobacterium odoratum* (*n* = 1), *Pseudomonas aeruginosa* (*n* = 4), *Pseudomonas oryzihabitans* (*n* = 1), and *Stenotrophomonas maltophilia* (*n* = 18).

Based on EUCAST guidelines [30], minimal inhibitory concentration (MIC) for sensitivity to meropenem (MEM), cefotaxime (CAZ), and ceftazidime (CTX) could be assessed for *Enterobacterales, Pseudomonas* spp. and *Acinetobacter* spp., because specific breakpoints were available. To differentiate between sensitive and resistant isolates, quantitative MICs were converted into binary qualitative values (sensitive/resistant) using the cut-off values provided by EUCAST [30]. A third category (sensitive, increased exposure) was recently introduced, in substitution of the former “intermediate” category, referring to the likelihood of therapeutic success when the dosing regimen of the antimicrobial or its concentration at the site of infection is increased. In addition, the epidemiological cut-off (ECOFF) values were used to distinguish between microorganisms with and without acquired resistance mechanisms. Conversely, for bacterial species not included in the EUCAST guide [30], only quantitative MICs values to MEM, CTX, and CAZ were recorded.

As shown in Table 1 and Table 2, resistant values to MEM were observed in strains of *P. aeruginosa* and *P. oryzihabitans*. In addition, one *E. coli* and one *E. americana* strain showed MICs for MEM higher than the ECOFF (value (>0.125 μg/mL).

Among *Enterobacterales*, resistance to CAZ was detected in 50% of the isolates (4/8; *E. agglomerans*, *E. coli,* and *E. americana*) and resistance to CTX in 62.5% of the isolates (5/8; *C. freundii*, *E. agglomerans*, *E. coli*, and *E. americana*) (Table 1). Resistance to CAZ was detected for *P. oryzihabitans*, while *P. aeruginosa* isolates were categorised as “sensitive, increased exposure”. MIC breakpoints for CAZ and CTX were not available for the other species found in pig faeces.

### 2.2. Phenotypical Antimicrobial Resistance in Human Isolates

The only bacterial species shared by humans and pigs was *P. aeruginosa*. Two isolates were detected from the human clinical samples (2/187; 1.1%). One strain (NEF 23) was resistant to MEM (MIC_MEM_ 16 μg/mL), and the other (NEF 156) was classified as “sensitive, increased exposure” (MIC_MEM_ 8 μg/mL). The two isolates were considered “sensitive, increased exposure” to CAZ (MIC_CAZ_ 4 μg/mL). Even if the sensitivity to CTX could not be evaluated, due to the absence of the clinical breakpoint, the MIC values were high for both strains (128 μg/mL and 64 μg/mL, respectively).

### 2.3. Whole-Genome Sequencing (WGS) of Porcine and Human Isolates

According to the aims of the present study, *P. aeruginosa* isolates from pigs and patients were sequenced by using MiSeq Illumina Platform (Illumina, Milan, Italy). In pigs, the carriers of four CR *P. aeruginosa* strains (apparent prevalence 1.3%; 95% CI 0.5–3.4) were reared on four farms in the Lombardy Region, northern Italy. As shown in Table 2, *P. aeruginosa* strains belonged to four sequence types—namely, ST274, ST782, ST885, and ST938. All of the isolates harboured the *bla*_PAO_ gene. In addition, *bla*_OXA-50_ (ST782 and ST885), *bla*_OXA-396_ (ST938), and *bla*_OXA-486_ (ST274) genes were found, together with genes for resistance to aminoglycosides *(aph(3′)-IIb*), chloramphenicol (*catB7*), ciprofloxacin (*crpP*), and fosfomycin (*fosA*).

Results on de novo assembly statistics are reported in Table 3 and Appendix A. Genome sizes (from 6,396,234 to 7,099,705 bp) and GC content (from 65.85% to 66.83%) confirmed the *P. aeruginosa* species identification of the newly sequenced genomes [31]. The number of contigs ranged from 124 to 317, N50 (the length of the shortest contig of the 50% of the assembly including equal or longer contigs) ranged from 38,674 to 126,411, and the maximum contig length from 223,278 to 561,477. These values highlight the high quality of de novo assemblies in terms of contiguity as reported previously on both *P. aeruginosa* genomes and bacterial genomes of similar size [32,33]. The sequencing coverage is a parameter related to the number of times each nucleotide of a given genome is sequenced starting from the assumption that reads are randomly distributed. The average depth of sequencing coverage is equal to LN/G, where L and N are the length and the number of reads, respectively, and G is the length of the haploid genome [34]. A coverage higher than 50× is generally recommended to allow reliable calling of single nucleotide variants (SNVs) such as single nucleotide polymorphisms (SNPs). In the present study, the coverage ranged from 76.6 to 105.2×.

SNP calling confirmed results of MLST typing with a higher level of discriminatory power (Figure 1A). Based on the maximum likelihood phylogenetic tree built on 68790 SNPs, newly sequenced genomes were distantly related except for ST938 isolates CRE 102 and NEF 23 of porcine and human origin, respectively. *P. aeruginosa* genomes CRE 153, CRE 295, CRE 98, and NEF 156 showed pairwise SNPs distances ranging from 25,928 to 33,748, whereas CRE 102 and NEF 23 showed 86 SNPs of difference confirming their genetic relationship (Appendix A). NEF 23 was collected from an 83-year-old male patient suffering from an episode of acute kidney insufficiency due to sepsis, on previous chronic kidney disease. Not surprisingly, comparing genomes of different ST-types (ST274, ST782, ST885, ST395) the genetic distance increased with pairwise SNPs differences (Appendix A). The genetic relatedness between CRE 102 and NEF 23 was reinforced by cgMLST results, which identified 661 allelic differences corresponding to the 25% on a total of 2653 alleles of the cgMLST scheme (Figure 1B). The other genomes showed allelic differences ranging from 2133 to 2219 corresponding on average to 82% of the scheme.

As shown in Table 2, the porcine and human isolates were characterised by an identical resistome consisting of five antimicrobial genes conferring decreased susceptibility to β-lactams and meropenem (*bla*_OXA-396_, *bla*_PAO_) as well as resistance to aminoglycosides (*aph(3′)-IIb*), chloramphenicol (*catB7*), and fosfomycin (*fosA*). Epidemiological links between the carrier pig (reared in Cremona Province, Lombardy Region) and the patient (living in Parma Province, Emilia–Romagna Region) were not investigated because of the lack of information about the patient’s eating habits, pig exposure, or living environment.

The other *P. aeruginosa* strain (NEF 156) isolated from a 52-year-old female patient suffering from acute kidney insufficiency belonged to ST395 and harboured *bla*_PAO_ and *bla*_OXA-488_ genes, thus differing from the porcine isolates detected in the study. Its resistome consisted of six determinants conferring decreased susceptibility to β-lactams and carbapenems (*bla*_OXA-488_, *bla*_PAO_) and resistance to aminoglycosides (*aph(3′)-IIb*), chloramphenicol (*catB7*), ciprofloxacin (*crpP*), and fosfomycin (*fosA*) (Table 2).

### 2.4. Localisation of Carbapenemase-Encoding Genes

Based on RAST and PROKKA annotation results and confirming previous data, *bla*_OXA_ and *bla*_PAO_ genes showed genetic environments which did not show any specific sequence of mobilisable genetic elements (MGEs) (Figure 2 and Figure 3).

### 2.5. Detection of Carbapenemases, β-Lactamases, and AmpC Genes in Other Pig Isolates

PCR testing of the strains detected in pigs revealed negative results for *bla*_KPC_, *bla*_IMP_, *bla*_VIM,_ and *bla*_NDM_ genes. *bla*_TEM-1_ and *bla*_CTX-M1_ were found in 100% and 26.3% (10/38) of the isolates, respectively. *bla*_SHV_ was detected in 5.3% (2/38) of the isolates, and precisely in one *E. coli* strain resistant to CTX and CAZ and in one *B. cepacia* showing high MIC values for CTX and CAZ. Among plasmidic AmpC genes, *bla*_MOX_, *bla*_DHA,_ and *bla*_FOX_ were never detected. *bla*_CIT_ was found in 34.2% (13/38) of the isolates and exactly in a MEM-sensitive *C. freundii* and in 12 *S. maltophilia*. *bla*_ACC_ and *bla*_EBC_ were found in one (2.6%) and two (5.3%) *S. maltophilia* isolates, respectively (Table 1).

## 3. Discussion

In recent years, the concept of One Health has been reformulated underlying the role of geographically close ecosystems in the occurrence of traits which have an impact on the human, animal, plant, and environmental health, such as AMR. Therefore, the studies of CR in microorganisms shared by food-producing animals and humans are crucial to assessing the contribution of animals to CIA resistance transmission. AMR bacteria in food-producing animals can be transmitted to humans via food-borne routes, as demonstrated for some zoonotic microorganisms (e.g., thermotolerant *Campylobacter*, *Salmonella*, Shiga toxin-producing *Escherichia coli*). At slaughter, carcass contamination by microorganisms shed by pigs can occur [35,36,37], thus causing their spread to consumers via ingestion of raw and undercooked pork and products thereof. Transmission of AMR bacteria from animals to humans may also occur through environmental contamination and direct animal contact. In addition, the commensal bacterial flora can act as a reservoir of AMR genes, which may be transferred to microorganisms capable of causing disease in both humans and animals [38].

### 3.1. CR P. aeruginosa in Pigs and Humans

Our study was focused on the detection of CR microorganisms in pigs entering the slaughter line, with a special interest in those bacterial species contemporarily found in patients hospitalised in the same geographical area. To the best of our knowledge, this is the first report of a CR *P. aeruginosa* ST938 strain carrying *bla*_OXA-396_ and *bla*_PAO_ genes isolated from pigs and humans in northern Italy (MIC_MEM_ 16 µg/mL). No epidemiological links were demonstrable between the animal (slaughtered in 2018) and the patient (hospitalised in 2019) so we could not hypothesise any role of pigs or pork in its transmission. In fact, anamnestic data collected from the patients did not include any information on their eating habits or professional/family association with pigs or the pork industry. Nevertheless, we could not exclude that, in some circumstances, pigs, pork, and products thereof could be a source of CR microorganism transmission to humans.

As is well known, *P. aeruginosa* is intrinsically resistant to most antimicrobials due to its ability to prevent penetration or to extrude different molecules from the cell by expressing several efflux pump systems and to the chromosomally encoded AmpC β-lactamase [39,40,41]. However, it can acquire resistance through chromosomal mutations as well as the acquisition of AMR genes via horizontal transfer [42]. Its ability to acquire resistance to antimicrobials and to adapt to different environmental conditions are the bases for its increasing prevalence in nosocomial infections worldwide [43,44]. The occurrence of CR *P. aeruginosa* in human patients varies in prevalence in the European countries, ranging from below 5% to more than 50% [45].

In pigs, other three isolates of *P. aeruginosa* carrying *bla*_PAO_, together with *bla*_OXA-50_, *bla*_OXA-486_, and *bla*_OXA-488_, genes were detected from animals reared on different farms. *bla*_OXA-50_ is an intrinsic oxacillinase, which can confer decreased susceptibility to ampicillin, ticarcillin, and meropenem [46]. Specifically, following the RAST annotation [47], the OXA-50 family comprises also the *bla*_OXA-396_, *bla*_OXA-486_, and *bla*_OXA-488_ genes. The MDR nature of *P. aeruginosa* isolates is assessed by the presence of resistance determinants to aminoglycosides, ciprofloxacin, and fosfomycin, which are critically important antimicrobials (CIAs) for human medicine, as well as resistance to chloramphenicol, which is a highly important antimicrobial in human treatments [48]. In parallel with *bla*_PAO_ and *bla*_OXA_ genes, genes with resistance to aminoglycosides, ciprofloxacin, chloramphenicol, and fosfomycin were also shown in the human *P. aeruginosa* ST938 and ST395 strains.

In most cases, *P. aeruginosa* does not have a primary role in pig infections but is frequently reported by laboratories because of its easiness of identification [49]. Although its role in the pathogenesis of atrophic rhinitis has been suggested, it seems to be unable to initiate disease in the nasal cavities of specific-pathogen-free pigs [50]. Some *P. aeruginosa* strains of porcine origin produce enterotoxins that involve fluid accumulation [51], while others can be involved in cystitis, vaginal infections, mastitis, and septicemias in newborn pigs caused by ascending infections [49]. In humans, *P. aeruginosa* is considered an opportunistic pathogen, causing severe infections in patients with impaired health status [52], and the emergence of CR *P. aeruginosa* is a significant contributor to patient morbidity and mortality [53].

### 3.2. ESBL/AmpC Genes Harbouring Porcine Isolates

The absence of clinical breakpoints and ECOFF values for most of the microorganisms detected in this study made it impossible to assess their resistance to carbapenems or deviation from “wild-type” phenotypes. For this reason, since their phenotypical resistance to carbapenems was only putative based on breakpoints set up for other species, all isolates were tested for carbapenemase genes. In addition, since decreased sensitivity to carbapenems could be caused by ESBL and AmpC β-lactamases associated with outer membrane porin loss [18,54,55], the isolates were tested both for phenotypical resistance to third-generation cephalosporins—commonly used in pig husbandry—and the presence of cephalosporinase encoding genes [56].

Among *Enterobacterales*, *E. coli* was the most common species detected in the study. Despite carbapenemase-producing genes having been detected in *E. coli* of porcine origin in different countries [15,57,58,59,60], our isolates lacked *bla*_KPC_, *bla*_IMP_, *bla*_VIM,_ and *bla*_NDM_ genes. They harboured *bla*_TEM-1_ (100%), alone or in combination with *bla*_CTX-M1_ (25%) or *bla*_SHV_ (25%). Our findings suggest a different distribution of ESBL genes from those reported by Rega et al. [61], which detected *bla*_CTX-M1_ in 89% and *bla*_TEM-1_ in 57% of *E. coli* strains of pork origin. MICs for meropenem were lower than the ECOFF in all the isolates, except in one strain (4 µg/mL) harbouring *bla*_TEM-1_. The same result was obtained for an isolate of *E. americana* (MIC_MEM_ 4 µg/mL), which was resistant to CAZ (MIC_CAZ_ 32 µg/mL) and CTX (MIC_CTX_ 16 µg/mL) and harboured *bla*_TEM-1_ and *bla*_CTX-M1_ genes. Interestingly, since *E. americana* is a rare but true human pathogen, able to cause serious invasive infections such as meningitis [62,63], its carriage by pigs is of concern. Globally, the contemporary presence of *bla*_TEM-1_ and *bla*_CTX-M1_, or *bla*_TEM-1_ and *bla*_SHV_, in members of the *Enterobacterales* was always associated with resistance to CAZ and CTX, while detection of only *bla*_TEM-1_ (100% of the isolates) was not.

Among *Pseudomonas* spp., *P. oryzihabitans* showed resistance to MEM (256 µg/mL) as well as to CAZ (256 µg/mL) and harboured *bla*_TEM-1_. This species is rarely isolated both from pigs and humans; regarding the latter, it can cause bacteriemia and peritonitis in immunocompromised patients [64,65].

The only species belonging to *Acinetobacter* spp. found in pig faeces was *A. lwoffii*. The isolate was sensitive to MEM according to the MIC testing and carried *bla*_TEM-1_, but its sensitivity to CAZ and CTX could not be evaluated for the absence of MIC breakpoints.

*bla*_TEM-1_ and *bla*_CTX-M1_ genes were detected in other species, such as *A. hydrophila, B. cepacia*, *C. indologenes*, and *F. odoratum*. Overall, MICs for MEM ranged from 4 µg/mL to 16 µg/mL, MICs for CAZ from 8 µg/mL to 512 µg/mL, and MICs for CTX varied between 16 µg/mL and 256 µg/mL. No relationship between cephalosporinase genes and level of resistance was found, since *B. cepacia* harbouring *bla*_TEM-1_, *bla*_CTXM-1_, and *bla*_SHV_ showed lower MIC values than *C. indologenes* carrying only *bla*_TEM-1_. These findings suggest that detection of cephalosporinase genes alone could not be indicative of resistance since other mechanisms are interrelated and contribute to AMR. 

Interestingly, *B. cepacia* and *C. indologenes* are human opportunistic pathogens in immunocompromised patients [66,67], and their carriage by pigs is of concern. Another isolate carrying *bla*_TEM-1_ and *bla*_CTX-M1_ was *A. hydrophila*, which can cause diarrhoea in piglets [68], and is resistant to a wide range of antimicrobials [69] and an opportunist pathogen in immunocompromised people [70]. Concerning *bla*_TEM-1_-positive *F. odoratum, Flavobacterium* spp. are reported as nosocomial-acquired opportunistic pathogens, due to their occurrence in patients with advanced immunodeficiency and resistance to many antimicrobials [71].

Lastly, the most common species detected in pig faeces was *Stenotrophomonas maltophilia*, an environmental multidrug-resistant (MDR) microorganism which exhibits resistance to several classes of antimicrobials, including β-lactams, cephalosporins, carbapenems, trimethoprim/sulfamethoxazole, macrolides, fluoroquinolones, aminoglycosides, chloramphenicol, tetracyclines, and polymyxins [72]. *S. maltophilia* is an opportunistic pathogen responsible for nosocomial infections, diseases in immunocompromised people, and chronic pulmonary infections in patients with cystic fibrosis [73,74]. Most *S. maltophilia* isolates (77.8%) carried *bla*_TEM-1_, and the remaining ones carried *bla*_TEM-1_ and *bla*_CTX-M1_. Their MICs values for MEM, CAZ, and CTX could not be interpreted as resistant or susceptible in absence of EUCAST clinical breakpoints.

Globally speaking, the emergence and distribution of CR Gram-negative pathogens, with a special interest in *Enterobacterales,* influences the outcome of human infections [75,76] because CR Gram-negative bacilli remain a threat, as few antimicrobial agents are reliably active.

## 4. Materials and Methods

### 4.1. Sample Collection

From January 2017 to March 2018, 300 finishing pigs were randomly selected during 23 visits to a slaughterhouse in Parma Province. The animals belonged to 300 batches and were reared on 95 farms in northern Italy, including the regions of Lombardy, Veneto, Emilia–Romagna, Piedmont, and Tuscany. The maximum distance between farms was 477 km, and the minimum distance was 3.5 km, with an average of 21 km. Most farms (72/95) were in the Lombardy Region, where pig husbandry has the highest density in the country. The faecal material was aseptically collected from the caecum of pigs immediately after evisceration and placed in sterile containers, stored at 4 °C, and transported to the laboratory on the day of sampling.

### 4.2. Sample Testing

The faecal samples were tested following the method proposed by the Technical University of Denmark [77] for the detection of ESBL-, AmpC-, and carbapenemase-producing *E. coli*, with some modifications. A 10 g aliquot of caecal material was suspended in 90 mL of buffered peptone water (BPW; Biolife Italiana, Milan, Italy) and incubated at 37 ± 1 °C for 18–22 h. After enrichment, cultures were tested as follows: (i) first, 100 µL were seeded onto MacConkey agar (Oxoid, Basingstoke, UK) plates aseptically added with a 10 µg meropenem disk (Oxoid), and then (ii) 10 µL were plated onto Brilliance CRE agar (Oxoid), a chromogenic selective medium which provides presumptive identification of CR microorganisms, such as *E. coli* (pale pink colonies), the *Klebsiella*, *Enterobacter*, *Serratia*, and *Citrobacter* (KESC) group (blue colonies) and *Acinetobacter* spp. (white to naturally pigmented colonies). Plates were incubated aerobically at 35 ± 2 °C for 18–24 h. Brilliance CRE negative plates were incubated for an additional 24 h before being discharged.

Each colony grown in Brilliance CRE agar plates was selected regardless of its colour and shape, and we considered both lactose-fermenting and non-fermenting colonies grown in the proximity of the meropenem disk in MacConkey agar plates. The selected colonies were subcultured onto trypticase soy agar plates (TSA; Oxoid). After incubation at 37 ± 1 °C for 18–22 h, pure cultures were Gram-stained. Gram-positive cultures were discharged.

### 4.3. Isolate Screening and Species Identification

Gram-negative isolates were tested for oxidase activity and screened for resistance to MEM using the Kirby–Bauer disk diffusion susceptibility test following the European Committee on Antimicrobial Susceptibility Testing (EUCAST) recommendations [78]. Zone inhibition diameters of oxidase-negative strains were evaluated on the bases of the clinical and ECOFF breakpoints proposed for the *Enterobacterales* [79]. For oxidase-positive isolates, the clinical breakpoints for *Pseudomonas* were followed, as they were the only ones available. Resistant isolates or isolates with zone diameters smaller than the ECOFF value (available for *Enterobacterales* only) were identified at the species level by using API^®^ 20 E system (bioMérieux, Marcy l’Etoile, France) or API^®^ 20 NE system (bioMérieux). After identification, the isolates were frozen at −80 °C for further testing.

### 4.4. MIC Testing

The conserved cultures were revitalised via inoculation in non-selective BPW at 37 °C overnight. The minimal inhibitory concentration (MIC) test for sensitivity to MEM, CAZ, and CTX was carried out following EUCAST guidelines [80]. Briefly, 96-well plates were used to perform a 2-fold serial dilution of the antibiotics tested. The bacterial suspension was added at the final concentration of 5 × 10^5^ CFU mL^−1^, and each test was repeated 3 times. The MIC values were determined by wide-eye reading. Results were recorded as the lowest concentration of antimicrobial agent that completely inhibits visible bacterial growth expressed in μg/mL. *E. coli* ATCC 25922 (ATCC, Manassas, VA, USA) was tested periodically as a quality-control microorganism.

Concerning *Enterobacterales* [81], both the MEM-resistance clinical breakpoint (>8 µg/mL) and the ECOFF value (>0.125 µg/mL) were available. For *Pseudomonas* spp. and *Acinetobacter* spp., MEM-resistance clinical breakpoints were different in case of meningitis (>2 μg/mL) or other clinical indications (>8 μg/mL). Following the purpose of the study, MIC > 8 μg/mL was considered indicative of resistance.

Even in absence of MIC breakpoints for most of the species detected in swine, all of the isolates were tested for sensitivity to MEM, CAZ, and CTX.

### 4.5. Comparison with CR Human Isolates

After performing the study on pigs, a survey on 187 patients affected by renal diseases hospitalised at the Parma University Hospital was conducted in 2019. Three hundred urine samples were collected from indwelling bladder catheters. Most patients (mean age 77 years (range 58–88)) were tested more than once because of prolonged hospitalisation and had been admitted with a diagnosis of acute kidney injury (97/187, 51.9%) or kidney transplant (90/187, 48.1%). The selection of patients was based on their willingness to participate in the study and their capacity to urinate. Urine aliquots of 1 mL were analysed following the protocol of the Center for Disease Control and Prevention [82], with some modifications. Following the purpose of the study, a comparison between the CR human isolates which belonged to the same species detected in pigs was performed by sequencing their entire genome.

### 4.6. Genotypic Confirmation Test

The isolates belonging to bacterial species not shared by pigs and humans were not analysed by WGS but were tested via PCR for carbapenemase-, ESBL-, and AmpC-producing genes.

A multiplex end-point PCR was used for the detection of *bla*_KPC_, *bla*_NDM_, *bla*_VIM_, *bla*_IMP,_ and *bla*_OXA-48-like_ genes. Five colonies of overnight bacterial culture on TSA (Oxoid) were diluted in 1 mL of sterile distilled water. DNA extraction was performed by heating at 95 °C for 10 min, and cellular debris was removed via 15,000 rpm centrifugation for 5 min. The supernatant was used for amplification after proper quantification with Biospectrometer Basic Eppendorf (Eppendorf, Milan, Italy). The primer sequences used are the ones described by Doyle et al. [83], as well as those indicated in the multiplex PCR protocol, with some modifications. The amplification was carried out with a GoTaq G2 Flexi DNA Polymerase Kit (Promega Italia, Milan, Italy). The master mix was prepared for 50 μL of final volume reaction containing 5× Green GoTaq Flexi Buffer at a final concentration of 1×, 2 mM of MgCl_2_, 0.2 mM of dNTPs, and 2 U of GoTaq G2 Flexi DNA Polymerase. *bla*_KPC_, *bla*_VIM_, and *bla*_IMP_ primers were added at a final concentration of 0.3 μM, *bla*_NDM_ at 0.4 μM, and *bla*_OXA-48-like_ at 0.5 μM. A total of 1 μL of sample lysate was added to the reaction mixture and nuclease-free water to the final volume. The PCR protocol consists of an initial denaturation of 95 °C for 5 min, followed by 35 cycles of DNA denaturation at 95 °C for 45 s, primer annealing at 62 °C for 45 s, and extension at 72 °C for 1 min. The final extension was performed at 72 °C for 8 min. PCR products were evaluated by electrophoresis with 1.5% agarose gels stained with SYBR Safe DNA gel stain (Invitrogen, Carlsbad, CA, USA) and visualised via UV light. A 100 bp DNA ladder from Promega (Milan, Italy) was used as a marker. Positive, negative, and no-template controls were included.

The ESBL genes *bla*_CTX-M1_, *bla*_CTX-M2_, *bla*_TEM-1_, and *bla*_SHV_ were detected following the real-time PCR protocol described by Roschansky et al. [84], with some modifications. DNA extraction was performed as previously described. Real-time amplifications were performed in 20 μL reactions containing 10 μL Gotaq qPCR Mix 2× (Promega Italia) at a final concentration of 1×. Forward and reverse primers were added at a final concentration of 0.3 μM. Each gene was tested individually. Supplemental CXR reference dye was added at 300 nM. A total of 1 μL of sample lysate was added to the reaction mixture and nuclease-free water to the final volume. The amplification protocol included a denaturation step (95 °C for 3 min) and 39 repeated cycles (95 °C for 15 s; 50 °C for 15 s; and 72 °C for 20 s). Fluorescence signals were collected in every cycle, and each sample was tested twice. Positive, negative, and no-template controls were included.

AmpC resistance genes (*bla*_MOX_, *bla*_CIT_, *bla*_DHA_, *bla*_ACC_, *bla*_EBC_, and *bla*_FOX_) were tested following the multiplex PCR protocol described by Pérez-Pérez and Hanson [85], with some modifications. The multiplex PCR master mix was prepared as described for carbapenemases, with the exception of primer final concentration (*bla*_MOX_, *bla*_CIT_, and *bla*_DHA_ at concentrations of 0.6 μM, *bla*_ACC_ and *bla*_EBC_ at 0.5 μM, and *bla*_FOX_ at 0.4μM) and 1.25 U of GoTaq G2 Flexi DNA Polymerase. The PCR amplification protocol consisted of an initial denaturation of 94 °C for 3 min, followed by 25 cycles of DNA denaturation at 94 °C for 30 s, primer annealing at 64 °C for 30 s, and extension at 72 °C for 1 min. The final extension was performed at 72 °C for 7 min. PCR products were evaluated via electrophoresis with 2% agarose gels stained with SYBR Safe DNA gel stain (Invitrogen) and visualised via UV light. A 100 bp DNA ladder from Promega was used as a marker. Positive, negative, and no-template controls were included.

### 4.7. Whole-Genome Sequencing

Whole-genomic DNA of the porcine and human *P. aeruginosa* isolates was extracted using the MagAttract HMW DNA Kit (QIAGEN, Hilden, Germany). The purified DNA concentration and the quality parameter ratio 260/280 were measured via BioSpectrometer fluorescence (Eppendorf). Whole genomes were sequenced on the Illumina MiSeq platform (Nextera library, paired-end reads). The INNUca v3.2 pipeline was used to quality check and de novo assemble reads into contigs [86]. The pipeline includes SPAdes v3.11 as a de novo assembler and provides multilocus sequence typing (MLST) profiles. Standard descriptive statistics of contig sequences were estimated with contig_info v2.01 [87].

SNP calling was performed on reads using snippy v4.0.5 pipeline with default settings and *Pseudomonas aeruginosa* PAO1 (NCBI Ref Seq NC_002516.2) was selected as reference genomes [88]. The reference was selected based on a high nucleotide-level genomic similarity (>99% of average nucleotide identity, ANI) in pairwise comparison with other *P. aeruginosa* genomes, assessed with FastANI v1.33 [89]. A maximum likelihood (ML) phylogeny was inferred on the core SNP alignments using RAxML v8.2.12, applying the GTRGAMMA evolutionary model [90]. Branch support was estimated using 1000 bootstrap replicates. The resulting ML tree was visualised using iTOL v4.2.3 software [91]. Furthermore, a cgMLST analysis was carried out with the chewBBACA suite v2.8.5 [92] using an available 2653 loci scheme [93]. A minimum spanning tree was obtained with the Phyloviz online software [94], based on *P. aeruginosa* cgMLST allelic profiles using the implemented eBURST algorithm.

Analysis of resistome of genomes was performed using ABRicate [95]. With this tool, a BLAST search of genes included in the Resfinder database was performed on de novo assemblies [96]. De novo assemblies were annotated using Prokka v1.13.3 [97] and RAST version 2.0 [47]. Gbk files were used as input files for visualisation of the genetic environment of carbapenemase-encoding genes by SnapGene [98]. Assembled sequences are available at the National Center for Biotechnology Information [99] BioProject PRJNA587603.

## 5. Conclusions

In our study, an MDR *P. aeruginosa* ST938 strain carrying *bla*_OXA-396_ and *bla*_PAO_ was the only CP microorganism shared by pigs and humans. An epidemiological investigation was not conducted because the anamnestic data collected from the *P. aeruginosa* ST938—positive patient at the hospital did not include any information on his eating habits or professional/familiar personal association with pigs or the pork industry. Nevertheless, our finding suggests that swine and their derivative products probably play minor roles in the transmission of CP bacteria to humans in the area of the study.

The absence of clinical breakpoints for most of the bacterial species detected in pigs hampered the evaluation of their phenotypical resistance to carbapenems, thus requiring genomic testing. Although carbapenemase-producing genes were not found in microorganisms relevant to human health, such as *E. coli*, *Ewingella americana,* and *Stenotrophomonas maltophilia*, a wide range of ESBL- and AmpC-producing genes were detected. Such findings are of concern because, in case of porin deficiencies, expression of ESBL- and AmpC-producing genes could be co-responsible for resistance to carbapenems.

Another interesting finding of the study in pigs is their carriage of AMR human opportunistic pathogens (*Stenotrophomonas maltophilia*, *Burkolderia cepacia*, *Criseobacterium indologenes*, *Aeromonas hydrophila*, and *Flavobacterium*
*odoratum)*, as well as the carriage of a true human pathogen (*Ewingella americana)*. These data could shed new light on the possible transmission of bacterial pathogens between pigs and humans, thus improving our knowledge of the epidemiology of some human infections.

## Figures and Tables

**Figure 1 antibiotics-11-00777-f001:**
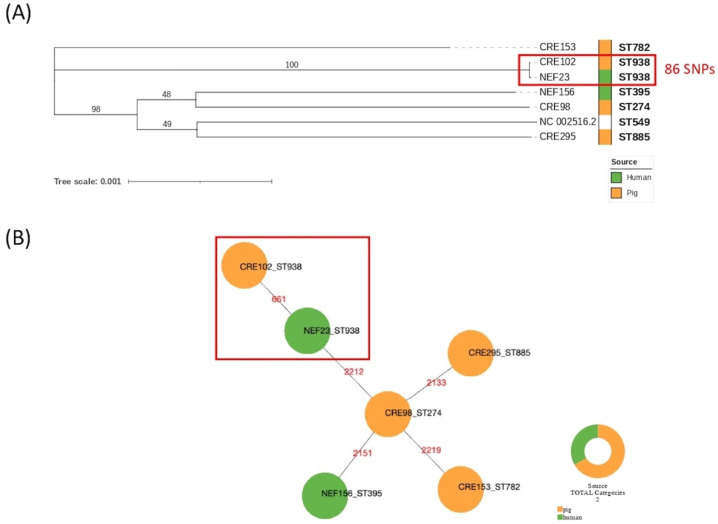
Phylogenetic trees showing genetic relationships among *P. aeruginosa* genomes included in this study: (**A**) core SNPs maximum likelihood tree (GTRGAMMA model) including the reference *P. aeruginosa* PAO1 (Ref Seq NC_002516.2), with bootstrap support labelled above branches; (**B**) cgMLST minimum spanning tree, in red the number of allelic variants.

**Figure 2 antibiotics-11-00777-f002:**
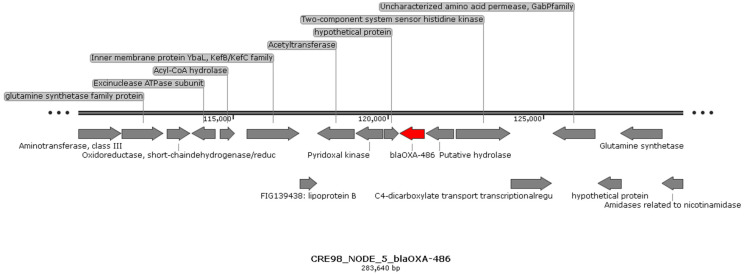
Genetic environment of *bla*_OXA-486_ gene in CRE 98 (ST274). The same genetic environment was observed for all detected genes belonging to the OXA-50 family in NEF 23 (ST938), CRE 102 (ST938), CRE 153 (ST782), CRE 295 (ST885), and NEF 156 (ST395) genomes.

**Figure 3 antibiotics-11-00777-f003:**
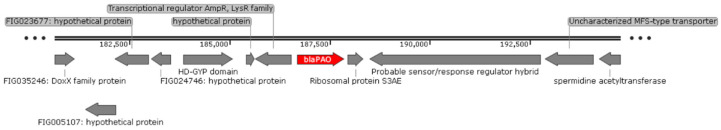
Genetic environment of *bla*_PAO_ gene in CRE 98 (ST274) genome. The same genetic environment was observed in NEF 23 (ST938), CRE 102 (ST938), CRE 153 (ST782), CRE 295 (ST885), and NEF 156 (ST395) genomes.

**Table 1 antibiotics-11-00777-t001:** MIC values for meropenem (MEM), cefotaxime (CTX), ceftazidime (CAZ), and resistant genes tested with PCR. MICs indicative of resistance for *Enterobacterales*, *Pseudomonas* spp., and *Acinetobacter* spp. are shown in bold. MICs higher than the meropenem screening breakpoint (ECOFF) are shown in Italics.

Species	MIC Values (Lg/mL)			*bla* Genes	*ampC* Genes
	MEM	CAZ	*CTX*		
*Enterobacterales*					
*C. freundii*	0.016	2	8	TEM-1	CIT
*E. agglomerans*	0.032	**128**	**512**	CTX-M-1, TEM-1	
*E. coli*	*4*	≤0.5	≤0.5	TEM-1	
*E. coli*	0.016	0.5	0.5	TEM-1	
*E. coli*	0.016	**8**	**4**	CTX-M-1, TEM-1	
*E. coli*	0.016	**32**	**512**	TEM-1, SHV	
*E. americana*	*4*	**32**	**16**	CTX-M-1, TEM-1	
*Pseudomonas* (no breakpoints for CTX)					
*P. oryzihabitans*	**256**	**256**	32	TEM-1	
*Acinetobacter* (no breakpoints for CAZ and CTX)					
*A. lwoffii*	0.016	0.5	8	TEM-1	
Species with no breakpoints					
*A. hydrophila*	4	32	64	CTX-M-1, TEM-1	
*A. hydrophila*	4	2	0.5	TEM-1	
*B. cepacia*	4	8	16	TEM-1	
*B. cepacia*	8	16	64	CTX-M-1, TEM-1, SHV	
*B. cepacia*	8	16	64	CTX-M-1, TEM-1	
*C. indologenes*	16	512	256	TEM-1	
*F. odoratum*	16	16	64	TEM-1	
*S. maltophilia*	128	64	256	TEM-1	
*S. maltophilia*	64	32	256	TEM-1	EBC
*S. maltophilia*	64	8	64	TEM-1	EBC
*S. maltophilia*	64	64	256	TEM-1	ACC
*S. maltophilia*	64	64	64	TEM-1	CIT
*S. maltophilia*	64	128	128	CTX-M-1, TEM-1	CIT
*S. maltophilia*	64	128	256	TEM-1	CIT
*S. maltophilia*	32	16	256	TEM-1	CIT
*S. maltophilia*	32	64	256	CTX-M-1, TEM-1	CIT
*S. maltophilia*	32	128	256	CTX-M-1, TEM-1	CIT
*S. maltophilia*	32	64	256	CTX-M-1, TEM-1	CIT
*S. maltophilia*	128	64	256	TEM-1	CIT
*S. maltophilia*	128	4	64	TEM-1	CIT
*S. maltophilia*	16	16	64	TEM-1	CIT
*S. maltophilia*	16	128	512	TEM-1	CIT
*S. maltophilia*	8	64	128	TEM-1	
*S. maltophilia*	4	2	16	TEM-1	CIT
*S. maltophilia*	1	2	128	TEM-1	

**Table 2 antibiotics-11-00777-t002:** Results from MIC testing and WGS of *P. aeruginosa* strains isolated from porcine and human sources. MICs indicative of resistance to meropenem (MEM) are in bold (EUCAST, 2021).

Source	ID Code	MIC Values (µg/mL)	Multi Locus Sequence Typing	*bla* Genes	Additional Resistance Genes
		MEM			
pig	CRE 98	2	274	OXA-486, PAO	*aph(3′)-IIb*, *crpP*, *fosA4, catB7*
pig	CRE 102	**16**	938	OXA-396, PAO	*aph(3′)-IIb*, *fosA4*, *catB7*
pig	CRE 153	**16**	782	OXA-50, PAO	*aph(3′)-IIb*, *crpP*, *fosA4*, *catB7*
pig	CRE 295	2	885	OXA-50, PAO	*aph(3′)-IIb*, *crpP*, *fosA4, catB7*
human	NEF 23	**16**	938	OXA-396, PAO	*aph(3′)-IIb*, *fosA4*, *catB7*
human	NEF 156	8	395	OXA-488, PAO	*aph(3′)-IIb, crpP*, *fosA4*, *catB7*

**Table 3 antibiotics-11-00777-t003:** Assembly statistics of sequenced genomes of *P. aeruginosa*.

Assembly ID	Source	Genome Size	GC%	No. of Contig	Coverage	N50	Maximum Contig Length
NEF 23	human	6,436,450	66.40	201	92.5	79,626	377,949
CRE 98	pig	6,415,664	66.40	124	99.8	126,411	561,477
CRE 102	pig	6,438,088	66.40	225	76.6	75,398	238,923
CRE 153	pig	6,396,234	66.43	317	79.2	38,674	223,278
CRE 295	pig	6,426,083	66.35	141	105.2	102,638	328,392
NEF 156	human	7,099,705	65.85	172	85.9	123,256	340,617

## Data Availability

Assembled sequences are available at the National Center for Biotechnology Information (NCBI) BioProject PRJNA587603.

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
