# Peer review of "Survey on Carbapenem-Resistant Bacteria in Pigs at Slaughter and Comparison with Human Clinical Isolates in Italy"

_antibiotics, 2022, doi:10.3390/antibiotics11060777_

Round 1
Reviewer 1 Report
The manuscript is suitable for publication in Antibiotics, but the language should be revised and additional analyses should be performed to present a comprehensive analysis as proposed by the authors.
I provide some major comments below.
Line 32 In how many? Line 67 Abbreviate Carbapenem-resistance and homogenize throughout the text. Line 84-89 Rewrite the text again, it is difficult to understand the author's idea. Line 99-101 Because the authors do not mention the use of cefotaxime and ceftazidime at the beginning, I do not understand the idea of adding it as additional. Line 106-120 The scientific names must be in italics. Revise the entire manuscript. Line 143: were "sequenced" by (e.g. Ion, Illumina, other) Figure 1. - Do not show the Boostrap number used. - Which strain is the "reference"? - What is the number of SNPs used to construct the phylogenetic tree? - CRE 102 and NEF 23 do not show any branches in the tree? Why include them if they do not contribute any information to the tree? - The scientific names must be in italics Table 3. They should include a column with the size of the sequenced genome and the source of the isolate. The authors have the complete genome of the isolates with few contigs and good coverage, the authors should complement the information presented with an analysis of the coregenome to reinforce the information reported with the SNPs, they have a large amount of information that is not taken advantage of. Line 309 What do the authors mean by "finishing pigs"? Line 432-433 concentra-tion? The results reported by RAST should be corroborated with prokka.Author Response
Dear Reviewer,
thank you for your valuable revision which could help us in improving our manuscript. We have followed all your suggestions in revising the text and additional analyses have been carried out. The changes in the text are highlighted in yellow.
Following your report, the “Introduction” section as well as the “Conclusions” section have been improved. New references have been added to the manuscript (Discussion section). The Maximum likelihood tree was improved, and additional analyses on the coregenome (cgMLST) as well as on the annotation (PROKKA) were performed. The language will be revised by a mother-tongue expert after final approval for publication.
Below you can find our answers to your questions:
Line 32 In how many? We have added the number of isolates harbouring ESBL and AmpC genes
Line 67 Abbreviate Carbapenem-resistance and homogenize throughout the text. Carbapenem-resistance was abbreviated here and throughout the text.
Line 84-89 Rewrite the text again, it is difficult to understand the author's idea. The text has been rewritten.
Line 99-101 Because the authors do not mention the use of cefotaxime and ceftazidime at the beginning, I do not understand the idea of adding it as additional. You are right; a sentence has been added to explain the importance of resistance to cefotaxime and ceftazidime
Line 106-120 The scientific names must be in italics. Revise the entire manuscript. All scientific names have been written in Italics. We are very sorry, it was an unvoluntary mistake due to copy-paste of the text in the journal template.
Line 143: were "sequenced" by (e.g. Ion, Illumina, other). Done as suggested: the words “sequenced by MiSeq Illumina platform” were added.
Figure 1. - Do not show the Boostrap number used. - Which strain is the "reference"? - What is the number of SNPs used to construct the phylogenetic tree? - CRE 102 and NEF 23 do not show any branches in the tree? Why include them if they do not contribute any information to the tree? - The scientific names must be in italics. Thank you for the comments. The Maximum Likelyhood phylogenetic tree was improved by adding the bootstrap values and rooting the tree on the reference which is P. aeruginosa PAO1 (RefSeq NC_002516.2) as reported in the Figure caption. By re-rooting the tree the genetic relationship between CRE102 and NEF 23 is more clear. In particular, we apologize for the mistake but 86 and not 41 SNPs differences were detected between the two genomes, underlining the relatedness but not the close relatedness of the genomes. Text has been changed accordingly. The number of SNPs used to build the tree was 68790. This value was added in the text “Based on the Maximum Likelihood phylogenetic tree built on 68790 SNPs, the ST938 isolates CRE102 and NEF23 of porcine and human origin.....”. Scientific names in the Figure caption are now in italics,
Table 3. They should include a column with the size of the sequenced genome and the source of the isolate.
The genome size was included along side the source of the isolate, N50, maximum contig lengh and GC%.
The authors have the complete genome of the isolates with few contigs and good coverage, the authors should complement the information presented with an analysis of the coregenome to reinforce the information reported with the SNPs, they have a large amount of information that is not taken advantage of. Thank you for the comment. We analysed the coregenome by cgMLST which reinforced SNPs calling outcomes. In particular, CRE102 and NEF23 were found as genetically related (although not closely related) differing by 661 alleles over 2653 alleles of the P. aeruginosa cgMLST scheme corresponding to 25% of allelic difference in comparison to more than 80% among the other genomes. The following sentence was added “The genetic relatedness between these two genomes was reinforced by cgMLST results which identified 661 allelic differences corresponding to the 25 % on a total of 2653 alleles of the cgMLST scheme”.
Line 309 What do the authors mean by "finishing pigs"? “Finishing pigs” are the pigs ready to be slaughtered. It’s a common term to describe pigs weighting 80 kg and more.
Line 432-433 concentra-tion? “Concentra-tion” was a mistake due to copy-paste of the text in the journal template. Now it is correctly written.
The results reported by RAST should be corroborated with prokka.
Thank you for your comment. We re-annotate the asseblies with PROKKA which confirmed the annotations of nodes 5 and 7 of CRE98 already done by RAST. In PROKKA a higher number of hypothetical proteins were assigned in comparison to RAST, therefore we decided to keep the RAST annotation in figures 2 and 3 of the manuscript. BlaOXA gene was annotated as blaOXA-50 by RAST and as blaOXA-133 by PROKKA. After comparison with the Resfinder database, the gene was annotated as blaOXA-486 which is gathered within the blaOXA-50 family. Similarly in NODE 7, the b-lactamase resistance associated gene was annotated as “class C beta- lactamase" by RAST and ampC (beta lactamase) by PROKKA. After comparison to the Resfinder database the gene was re-annotated as blaPAO. In the literature AmpC and blaPAO are two designations of the same gene.

Reviewer 2 Report
Introduction
This can be shortened by omitting some details, which are irrelevant to the study.
Materials and methods
4.1. If possible, please include a map with the locations of the 95 farms. If this is not possible, at least indicate the average distance between these farms.
4.5. Please indicate the possible associations of the patients with animals and pigs specifically: consumption of pork, professional association with pork industry, family association with pork industry, anything that will serve to extend the idea of one health within the context of the present work.
General. How were the carcasses and the patients included into the study, selected in the first place?
Results
2.1. Do you have data regarding the type of carcass or the part of carcass, from which each isolate was obtained?
Tables 1 and 2: too detailed, please transfer to supplementary material.
2.3. Table 3. The statistics of the assemblies are not acceptable. The authors must provide full details of the results.
Are there any real-life data to possibly associate CRE102 with NEF23?
Discussion
First. Please divide in two or three sub-sections for easier reading.
Second. Unless you have real-life data of association of the patient with the pig industry, please tone down the relevant comments.
Overall. An interesting manuscript that provides useful information and merits publication, after significant improvement as detailed above.
Author Response
Dear Reviewer,
Thank you for your valuable revision of our manuscript. All your comments and suggestions have received our greatest attention and any change thereof in the manuscript is highlighted in light blue.
The Introduction has been improved but we could not shorten it too much because this conflicted with other suggestions (the reviewers were four!). Anyway, we have shortened some sentences on ESBL/AmpC E. coli producers.
Our reply to your comments is given below.
Materials and methods
4.1. If possible, please include a map with the locations of the 95 farms. If this is not possible, at least indicate the average distance between these farms. Since the map describing the location of 95 farms was too complicated to include, we preferred including the average distance between farms.
4.5. Please indicate the possible associations of the patients with animals and pigs specifically: consumption of pork, professional association with pork industry, family association with pork industry, anything that will serve to extend the idea of one health within the context of the present work. You are right when you suggest that data of pork consumption, professional or family association with pork industry etc. could be of the greatest importance to the study. Unfortunately, those data were not collected from the hospitalised patients because they were not suffering from a foodborne disease, but a kidney disease. For foodborne diseases it is normal to ask what kind of food was eaten of any information about contact with food-producing animals, but for kidney diseases it is not.
General. How were the carcasses and the patients included into the study, selected in the first place? In our study we did not test carcasses, but faecal material collected from the pigs after slaughtering. The animals were randomly selected at slaughter during 23 visits, including pig batches from different Italian region. At the hospital, the patients were selected depending on their willingness to adhere to the study and their capacity to urinate (many patients suffered from anuria and therefore were out the study). I have added a sentence in 4.5 to describe the selection among patients.
Results
2.1. Do you have data regarding the type of carcass or the part of carcass, from which each isolate was obtained? We did not collected samples from pig carcasses, but faecal material from the caecum immediately after slaughter. Therefore, we have no data on carcasses.
Tables 1 and 2: too detailed, please transfer to supplementary material. Thank you for your suggestion. Some details were eliminated, such as ID codes of the isolates from Table 1, and MIC values for CTX and CAZ in Table 2. Nevertheless, Table 1 fully explains our results, all of which are necessary to understand the complexity of the study. Working without MIC breakpoints was very difficult, and Table 1 shows how to manage in their absence (PCR testing of the putative resistant isolates was the only solution).
2.3. Table 3. The statistics of the assemblies are not acceptable. The authors must provide full details of the results. Done as suggested. Additional data on N50, GC% and maximum contig length were included.
Are there any real-life data to possibly associate CRE102 with NEF23? Unfortunately, as discussed in the manuscript, no real-data are available to possibly associate CRE-102 (the pig isolate) with NEF-23 (the human isolate). No information on eating habits of the patients were available at the time of the study. In addition, testing of pigs and patients was carried out in different periods of time so any direct association between the shedding pig and the patient should be excluded.
Discussion
First. Please divide in two or three sub-sections for easier reading. The Discussion section has been divided into two sub-sections, as you suggested.
Second. Unless you have real-life data of association of the patient with the pig industry, please tone down the relevant comments. Comments on association of P.aeruginosa ST939 between pigs and humans have been toned down accordingly to your comment.
Reviewer 3 Report
Authors investigated resistance to carbapenems and cephalosporins in Gram-negative bacteria from pigs and urine samples of hospitalized patients. It was found meropenem-resistant Pseudomonas isolates and cefotaxime/ceftazidime-resistant E. coli, E. americana, etc. Meanwhile, ESBL and AmpC genes were existed in most isolates. carbapenem-resistant P. aeruginosa was shared in pigs and patients, and P. aeruginosa ST938 with blaPAO and blaOXA396 genes was detected in one pig and an old patient, indicating possible circulation of carbapenem-resistant bacteria between swine and humans. It is a very interest work and hopeful for preventing transmission of resistant bacteria. However, some questions need to resolve before publishing:
Line 110 and Table 1: the full name of MIC, MEM CAZ, CTX, etc.
Table 2: Source, full name of MLST, Additional…
Line 134-136: is there any other references between “sensitive” and “resistant” ones according to the MICs? Is it EUCAST guideline? Please add them if it is possible.
Line 350: what does mean thawing? Please give simple description of the MIC determination.
Line 353: Please give the source of E. coli ATCC 25922.
In conclusions section: Line 451-455 may be deleted, which may be remove to introduction or discussion.
Other minors:
Line 114, 319, 326, 431: italic bacterial name.
Line 194: Other
Line 394-396: 50 μl, 5× Green, MgCl2, 0.2 mM, etc.
References: Ref.11, 25, 52, 53, etc.
Author Response
Dear Reviewer,
Thank you for your valuable and clear revision of our manuscript. All your comments and suggestions have received our greatest attention and any change thereof is highlighted in green.
Line 110 and Table 1: the full name of MIC, MEM CAZ, CTX, etc. The full names of MIC, MEM, CTX and CAZ were written
Table 2: Source, full name of MLST, Additional… The changes have been made as suggested..
Line 134-136: is there any other references between “sensitive” and “resistant” ones according to the MICs? Is it EUCAST guideline? Please add them if it is possible. We have moved the explanation of “sensitive”, “resistant” and “sensitive, increased exposure” from the Material and Methods section to the Results section. The EUCAST guideline reference has been included.
Line 350: what does mean thawing? “thawing” means “defrost”. I have deleted the term and described the revitalizing procedure of the frozen cultures.
Please give simple description of the MIC determination. A description of the MIC method has been included in the text.
Line 353: Please give the source of E. coli ATCC 25922. The source of the strain has been given.
In conclusions section: Line 451-455 may be deleted, which may be remove to introduction or discussion. L 451-455 have been deleted from the Conclusions section and copied into the Discussion section.
Other minors:
Line 114, 319, 326, 431: italic bacterial name. Italic bacterial names have been written throughout the text. It was an unvoluntary mistake due to copy-paste of the manuscript in the journal template. This change is highlighted in yellow (not in green), because also another reviewer required it.
Line 194: Other. “Other” has been correctly written (capital letter)
Line 394-396: 50 μl, 5× Green, MgCl2, 0.2 mM, etc. Done as suggested, thank you.
References: Ref.11, 25, 52, 53, etc. We are very sorry but can’t understand how to correct the references.
Reviewer 4 Report
After the review of the submitted article, I consider that in the current wording it should be REJECTED, because it has serious flaws and additional experiments are needed, the research was not carried out correctly.
1.- The sampling is carried out (January 2017 to March 2018) before being approved by the Bioethics Committee (June 10).
2.- The 300 fecal samples are not representative of several Italian regions, since of the 95 farms the majority (76%) were in Lombardy.
3.- All enterobacteria should be written in Italics.
4.- In most of the enterobacteria strains, there was a lack of information in EUCAST of the reference MIC values.
5.- The relevant data are exclusively from 6 strains of Pseudomona aeruginosa isolated from 4 pigs and 2 people.
6.- Most of the information refers to the 18 strains Stenotrophomonas maltophilia, but their epidemiological importance is not indicated.
7.- Reference 44 should include OIE (World Organisation for Animal Health) name updated in 2005 of the International Organisation of Epizootics.
The final conclusion of the article is that the direct epidemiological link between carbapenem-resistant strains of Pseudomonas spp. circulating between pigs and humans could not be demonstrated.
Author Response
Dear Reviewer,
Thank you for the revision of our manuscript. Please find below our reply to all your comments. Changes in the text are highlighted in purple.
1.- The sampling is carried out (January 2017 to March 2018) before being approved by the Bioethics Committee (June 10). This was not true, because faecal sampling from slaughtered pigs needed no approval from the Bioethics Committees. Approval was needed for hospitalized patients’ sampling, and this was obtained in 2019 (the sampling in patients was performed in 2019, i.e. after the sampling in pigs)
2.- The 300 fecal samples are not representative of several Italian regions, since of the 95 farms the majority (76%) were in Lombardy. Selection of pigs from all the Italian region is not realistic in only one slaughterhouse, because pig farmers send their animals to the nearest one, thus avoiding long-lasting transportation of pigs. As a consequence, the 300 farms of the study were not representative of all the Italian regions but were representative of Northern Italy. The slaughterhouse and the hospital included in the study were in the Emilia-Romagna region, where the most important Italian pig slaughterhouses are located. On the contrary, the highest number of Italian pig farms are in the neighbouring Lombardy region, which consequently was the region with the highest number of tested farms. The random selection of 72 farms (of 95) from Lombardy was not a mistake but represented the real distribution of pigs sent to the slaughterhouse. We have added an explanation in the manuscript.
3.- All enterobacteria should be written in Italics. All Enterobacteria (and, in general, all bacterial species) have been written in Italics. It was an unvoluntary mistake due to copy-paste of the manuscript into the journal template and was observed by all the reviewers. Scientific names in Italics are highlighted in yellow.
4.- In most of the enterobacteria strains, there was a lack of information in EUCAST of the reference MIC values. Thank you for this comment, because it is really focused on one of the biggest problems we had during the study. Indeed, the lack of EUCAST MIC breakpoints for sensitivity/resistance to antimicrobials in some bacterial species was a complication of the study. We were not aware, at the beginning of the study, that most of the species isolated from pigs could have no MIC breakpoints, because the analyses were focused on the detection of Enterobacterales. No idea that Aeromonas, Stenotrophomonas, Flavibacterium, Burkholderia and others could be detected by using media for Enterobacterales. On our opinion, it is important to be aware that MIC values are not available for all bacterial species and that PCR testing could be the only solution to find out AMR isolates.
5.- The relevant data are exclusively from 6 strains of Pseudomona aeruginosa isolated from 4 pigs and 2 people. We are aware that the most relevant data are from 6 strains of P. aeruginosa, but another important information of the study is the detection in pigs of Ewingella americana and opportunistic human pathogens, such as S. maltophilia, , Burkholderia cepacia, C. indologenes, F. odoratum. Otherwise, carbapenenem-resistant P. aeruginosa is very common in Europe in hospitalized patients and deserves great attention.
6.- Most of the information refers to the 18 strains Stenotrophomonas maltophilia, but their epidemiological importance is not indicated. Your comment is correct, but the detection of Stenotrophomonas maltophilia in pigs needs more studies to assess the possible epidemiological role of swine in human infections. To date, information are very scarce and our study could be somewhat useful.
7.- Reference 44 should include OIE (World Organisation for Animal Health) name updated in 2005 of the International Organisation of Epizootics. Reference 44 has been updated, together with reference 8. Thank you for your suggestion.
The final conclusion of the article is that the direct epidemiological link between carbapenem-resistant strains of Pseudomonas spp. circulating between pigs and humans could not be demonstrated. As we have described in our manuscript, it is not easy to correlate the detection of pathogens in pigs and humans, when data from patients are not available. In Italy, information from patients about eating habits and contacts with food-producing animals are collected only in case of food-borne diseases. Th final conclusion of the paper is that in pigs we can isolate human pathogens harboring several AMR genes, including OXA, PAO, ESBL-and AmpC- genes. On our opinion, this is an important result to be shared with the scientific community.
Round 2
Reviewer 1 Report
The phylogenetic tree has basic inconsistencies.
1.- The boostrap value used is not indicated, and the values in the branches are not appreciated.
2.- It is not indicated which one (in the tree) and why it is the reference.
3.- It is not indicated the evolutionary model used.
It is mentioned that the genomes were annotated with RAST, but no evidence of the results is presented.
Author Response
Dear Reviewer,
thank you for your additional comments on the SNP based phylogenetic tree which could help us in improving our manuscript. Any changes in the text are highlighted in yellow. In particular, following your comments we improved the inference of the phylogenetic tree
Below you can find our answers to your questions:
The phylogenetic tree has basic inconsistencies.
1.- The boostrap value used is not indicated, and the values in the branches are not appreciated.
2.- It is not indicated which one (in the tree) and why it is the reference.
3.- It is not indicated the evolutionary model used.
It is mentioned that the genomes were annotated with RAST, but no evidence of the results is presented.
The bootstrap is now indicated as well as values in the branches are now more appreciable. A maximum likelihood (ML) phylogeny was inferred on the core SNP alignments using RAxML v8.2.12, applying the GTRGAMMA evolutionary model. Branch support was estimated using 100 bootstrap replicates. The resulting ML tree was visualized using iTOL v4.2.3 software. Details were added at L510-517.
The reference in the tree is now indicated. Regarding the choice of the reference, in case of homogenous dataset (genomes belonging to the same ST-type, we generally choose and internal genome as reference or a public one belonging to the same ST-Type (https://doi.org/10.1016/S2666-5247(21)00149-X ). In the case of the present study we have genomes belonging to different sources and different ST-Types. Therefore, we decided to use a public genome which was close to our genomes showing an average nucleotide identity (ANI) over 99% (L493-495).
|
Reference |
Sample |
ANI value |
Count of bidirectional fragment mappings |
Total query fragments |
|
Pseudomonas_aeruginosa_reference.fasta |
NEF-156_S20_L001.fasta |
99.2847 |
1969 |
2088 |
|
Pseudomonas_aeruginosa_reference.fasta |
CRE295_S12_L001.fasta |
99.2667 |
1962 |
2088 |
|
Pseudomonas_aeruginosa_reference.fasta |
CRE98_S10_L001.fasta |
99.2464 |
1976 |
2088 |
|
Pseudomonas_aeruginosa_reference.fasta |
CRE102_S11_L001.fasta |
99.1561 |
1933 |
2088 |
|
Pseudomonas_aeruginosa_reference.fasta |
NEF23_S7_L001.fasta |
99.1445 |
1944 |
2088 |
|
Pseudomonas_aeruginosa_reference.fasta |
CRE153_S9_L001.fasta |
99.0991 |
1830 |
2088 |
Regarding RAST, the annotation file was used to build figures 2 and 3 on the genetic environment of antimicrobial resistance associated genes as already reported in the M& M (L504-507). To reinforce this, we added a sentence also in the results section at L215.
Reviewer 2 Report
The presentation of the findings of WGS is disappointing. I do not know if the authors cannot or do not wish to present the data in detail, but this not really correct results of WGS. Sorry, but this is totally wrong.
I suggest that this is corrected or at least to write something along the lines of a brief summary of the full results of WGS.
A real disappointment in this particular context.
Author Response
Dear Reviewer,
we are pleased to answer to your comment that gives us the possibility to emphasise our WGS results. We did not do it before for a simple misunderstanding thinking the reviewer asked exclusively to add more parameters of quality in Table 3, for which we complemented in round 1 the output of quality check of de novo assemblies performed by INNUca pipeline to outputs of Contig_Info pipeline of the Institut Pasteur. For this round 2, we agree with the reviewer in his/her attempt to highlight important results such as those of WGS analyses, and we added relevant text and references.
Our reply to your comments is given below.
Round 1
2.3. Table 3. The statistics of the assemblies are not acceptable. The authors must provide full details of the results.
Round 2
The presentation of the findings of WGS is disappointing. I do not know if the authors cannot or do not wish to present the data in detail, but this not really correct results of WGS. Sorry, but this is totally wrong.
I suggest that this is corrected or at least to write something along the lines of a brief summary of the full results of WGS.
A real disappointment in this particular context.
We included a detailed paragraph fully describing Table 3 results (L179-L193). Moreover, two supplementary tables on quality statistics of de novo assembly and SNP distance matrix were included (Tables S1 and S2). We also described further, results on SNP calling (195-200) and cgMLST (L206-208). We hope this is now acceptable. If not, we kindly ask to the reviewer to be more specific in the WGS results he/she wants us to discuss in further details.
Reviewer 4 Report
The authors have justified all the suggestions for improvement and have transferred them to the text, highlighting the changes to facilitate their reading.
Author Response
Dear Reviewer,
we are very grateful for your efforts in reviewing our manuscript. Thank you very much for having appreciated our reply to all your comments.
Round 3
Reviewer 1 Report
The authors use a boostrap of 100 replicates, however, 1000 is the minimum number of replicates so that the value of the branches obtained has a more robust support.
English must be revised.
Author Response
Dear Reviewer,
Thank you for your suggestions. Our reply to your comments is given below.
Round 3
The authors use a boostrap of 100 replicates, however, 1000 is the minimum number of replicates so that the value of the branches obtained has a more robust support.
English must be revised.
Done as suggested: “100” was changed to “1000” in the materials and methods (L516). Figure 1A was substituted accordingly.
English will be revised after acceptance of the manuscript.